# Influence of the Sludge Retention Time on Membrane Fouling in an Anaerobic Membrane Bioreactor (AnMBR) Treating Lipid-Rich Dairy Wastewater

**DOI:** 10.3390/membranes12030262

**Published:** 2022-02-25

**Authors:** Maria Alejandra Szabo-Corbacho, Santiago Pacheco-Ruiz, Diana Míguez, Christine M. Hooijmans, Damir Brdjanovic, Hector A. García, Jules B. van Lier

**Affiliations:** 1Water Supply, Sanitation and Environmental Engineering Department, IHE Delft Institute for Water Education, Westvest 7, 2611 AX Delft, The Netherlands; t.hooijmans@un-ihe.org (C.M.H.); d.brdjanovic@un-ihe.org (D.B.); h.garcia@un-ihe.org (H.A.G.); j.b.vanlier@tudelft.nl (J.B.v.L.); 2Latitud—Fundación LATU, Laboratorio Tecnológico del Uruguay (LATU), Avda. Italia 6201, Montevideo CP 11500, Uruguay; dmiguez@latitud.org.uy; 3Department of Water Management, Sanitary Engineering Section, Faculty of Civil Engineering and Geosciences, Delft University of Technology, Stevinweg 1, 2628 CN Delft, The Netherlands; 4BIOTHANE—Veolia Water Technologies, Tanthofdreef 21, 2623 EW Delft, The Netherlands; santiago.pacheco-ruiz@veolia.com; 5Department of Biotechnology, Delft University of Technology, Van der Maasweg 9, 2629 HZ Delft, The Netherlands

**Keywords:** anaerobic membrane bioreactor, dairy wastewater, membrane fouling, sludge retention time, physicochemical characteristics

## Abstract

This study evaluated the effects of sludge retention time (SRT) on the membrane filtration performance of an anaerobic membrane bioreactor (AnMBR) fed lipid-rich synthetic dairy wastewater. The membrane filtration performance was evaluated in two AnMBR systems operated at two different SRTs, i.e., 20 and 40 days. For the AnMBR operated at 40 days, SRT exhibited worse membrane filtration performance characterized by operational transmembrane pressures (TMP) exceeding the maximum allowed value and high total resistances to filtration (R_total_). The sludge in the two reactors evaluated at the different SRTs showed similar sludge filterability properties. However, the sludge in the reactor operated at 40 days SRT was characterized by exhibiting the highest concentrations of: (i) total suspended solids (TSS), (ii) small-sized particles, (iii) extracellular polymeric substances (EPS), (iv) soluble microbial products (SMP), (v) fats, oils and grease (FOG), and (vi) long-chain fatty acids (LCFA). The cake layer resistance was the major contributor to the overall resistance to filtration. The high TSS concentration observed in the AnMBR systems apparently contributed to a less permeable cake layer introducing a negative effect on the membrane filtration performance.

## 1. Introduction

Dairy industry wastewater is characterized by high concentrations of organic matter, suspended solids, and fat, oil, and grease (FOG) compounds [1,2]. In addition, it contains a high concentration of lipids [3,4]. Anaerobic wastewater treatment is a suitable option for such wastewater, considering its capacity for processing high organic-loading rates at very low sludge production rates, while at the same time producing biogas (a gaseous energy carrier) [3]. The high FOG content in dairy wastewater can introduce operational limitations to the performance of conventional anaerobic wastewater treatment systems, including floating sludge and granule disintegration, among others [1]. However, those operational challenges eventually can be overcome by using membrane filtration as the solids-liquid separation process.

An anaerobic membrane bioreactor (AnMBR) combines the advantages of anaerobic biological processes with membrane filtration. AnMBR systems provide a complete retention of the biomass and allow for slow-growing microorganisms to be retained in the system when long sludge retention times (SRTs) are applied. Moreover, AnMBRs are characterized by a high treatment performance with high organic-matter removal efficiencies exceeding 98% [4]. In addition, the treated effluent has an excellent quality, i.e., low organic matter concentration and free of suspended solids, which is ideal for water reclamation applications [5]. Despite these advantages, the application of full-scale AnMBR systems is still limited due to their elevated capital and operational costs. AnMBR systems require additional equipment to operate compared to conventional systems, such as ultra-filtration membranes, pressure and level sensors, and chemicals for cleaning in place, among others. Moreover, the filtration process can be prone to membrane fouling, either increasing the trans-membrane pressure (TMP) of the filtration system for a given flowrate (or membrane flux) or decreasing the resulting flux for a given TMP. Despite the considerable decrease in the membrane costs in the past decades [6], the attainable membrane permeability, i.e., the permeate flux per unit of membrane area and applied TMP, remains the most important operational factor with a high impact on the overall capital and operational costs of AnMBR systems [2].

Long-chain fatty acids (LCFAs), i.e., the hydrolysis products of triacyl lipids, may have a negative impact on the anaerobic biomass activity. Such compounds may be adsorbed onto the biomass, eventually leading to mass-transfer limitations and to sludge flotation [1]. Moreover, these compounds may also affect the membrane filtration performance of AnMBR systems [3,4]. Only a few studies have reported the effects of the presence of lipids and LCFA on the sludge filterability and membrane filtration in AnMBRs. Dereli et al. [7,8] evaluated the filtration performance of an AnMBR treating corn-based bioethanol thin stillage wastewater (high lipids content) at different SRTs. The best membrane filtration performance was obtained at the lowest evaluated SRT of 20 days, which was attributed to the lower accumulation of lipids on the anaerobic biomass and fewer fine particles in the reactor. At 20 days SRT, the LCFA precipitated as fat balls and they were manually scooped off from the liquid bulk [7]. Diez et al. [9] evaluated the treatment performance of a submerged AnMBR treating a lipid-rich wastewater from a snack factory, FOG content ranging from 4.4 to 6 g/L, aiming at determining the optimum operational membrane flux and associated membrane chemical cleaning regimes. The authors reported relatively low long-term optimum operational fluxes ranging from 6.6 to 8 L/(m^2^ h) at an optimum filtration cycle of 11 min, including 10 s of pre-relaxation, 20 s of backwash and 70 s of post-relaxation. The addition of cleaning interventions with chemicals products and air scouring increased the membrane filtration performance. Finally, Carta-Escobar et al. [10] evaluated the membrane performance of an aerobic MBR system, treating lipid-rich dairy wastewater at organic loading rates, grams of chemical oxygen demand (COD) per liter per day, from 0.24 to 0.7 g COD/(L d). The authors reported severe membrane fouling associated with the production of a viscous film on the surface of the membranes. There have been a few attempts to evaluate the overall impact of lipid-rich wastewater on the membrane filtration performance and membrane fouling. Most of these studies reported a reduced membrane filtration performance when high concentrations of lipids were present in the influent wastewater.

The effects of specific sludge characteristics on the membrane filtration performance and membrane fouling have only been partially reported as follows. The present description includes the most relevant literature on this issue reported over the past years. Suspended solids can accumulate on the membrane surface forming a cake layer, contributing to add resistance to the membrane filtration process. In addition, several studies have reported on the negative effects of sticky substances secreted by the microbial biomass, i.e., extracellular polymeric substances (EPS) and soluble microbial products (SMP), on the membrane filtration performance [11,12]. In addition, other sludge characteristics, such as the particle size distribution (PSD), surface charge, and hydrophobicity can be linked to membrane permeability and membrane fouling [13,14,15]. For instance, previous studies reported strong correlations between the PSD of the sludge and membrane fouling [12]. The smaller the size of the particles present in the sludge mixture, the larger the impact on membrane fouling. It has been reported that the colloidal material can exert a significant role in membrane fouling by blocking the membrane pores [11].

The SRT is regarded as one of the most important operational parameters affecting both the biological processes and the sludge characteristics in an AnMBR system. High operational SRTs would result in high biomass concentrations inside the reactor, and possibly high biogas production rates [4,6]. However, high SRTs coincide with low sludge wastage, which may lead to the accumulation of substances such as lipids and LCFAs [7], inert matter, cell debris [16], SMP and EPS, with a potential negative impact on the membrane filtration performance [17,18].

The membrane filtration performance of an AnMBR system strongly depends on the interaction between the influent wastewater characteristics, the sludge properties, and the process operational conditions [6]. Both the process operational conditions and the influent wastewater composition determine the sludge physicochemical properties and thus the membrane filtration performance. There is a gap in the literature on assessing the membrane filtration performance of an AnMBR when treating lipid-rich wastewater (such as dairy wastewater), considering the effects of selected process operational conditions such as SRT, which had been reported as one of the most important parameters in AnMBRs [6]. The aim of this research was to evaluate the membrane filtration performance when treating lipid-rich wastewater from a dairy industry in an AnMBR at different SRTs. The effects of the dairy influent wastewater and operational SRTs on the sludge characteristics were determined, and the impact of the sludge features on the membrane filtration performance was assessed. To the best of the authors’ knowledge, this is the first study evaluating the impact of the SRT on the sludge characteristics and membrane filtration performance when treating lipid-rich dairy wastewater.

## 2. Materials and Methods

### 2.1. Experimental Design

The membrane filtration performance of two 10 L laboratory-scale AnMBR systems was evaluated. One reactor was operated at 20 days SRT, while the other was operated at 40 days SRT. Both reactors were fed with synthetic dairy wastewater. The OLR was gradually increased until reaching an OLR of 4.7 ± 0.8 g/(L d) for both reactors. The reactors were operated for 189 days. The impact of the sludge properties, i.e., TSS, dynamic viscosity, PSD, CST, SRF, SF, EPS, and SMP, on the membrane filtration performance was determined. The membrane filtration performance was assessed by measuring the flux decline and TMP, and by determining the total resistance to filtration defined as the ratio between the TMP and the dynamic viscosity and membrane flux. Other AnMBR operational parameters such as the HRT, temperature, and organic loading rate (OLR), among others remained unmodified; thus, only the effects of SRT on the membrane filtration performance could be assessed. SRT values of 20 and 40 days were chosen following the recommendations from the AnMBR manufacturer (Biothane, Veolia Water Technologies, Delft, The Netherlands).

### 2.2. Experimental Setup

The experimental setup consisted of two AnMBRs. Each system consisted of a continuously mixed 10 L jacketed glass reactor with a working volume of approximately 10 L, equipped with PVDF cross-flow tubular ultrafiltration membranes (Pentair X-Flow, X-Flow BV, Enschede, The Netherlands). Each membrane had a surface area of 0.049 m^2^. A PLC system was included to monitor and control the operation of the reactors. The TMP in the two systems was monitored by incorporating pressure transmitters (Series 33X, KELLER, Reeuwij, The Netherlands). The setups were provided with a permeate collection tank of 5 L made of jacketed glass. The influent wastewater was added to the AnMBR systems by a peristaltic pump (Watson-Marlow 1200s, Thermo Fisher Scientific Göteborg, Sweden) from a feed vessel of 20 L. The vessel was continuously mixed using a magnetic stirrer (IKA, RCT basic, KA^®^-Werke GmbH & Co. KG, Staufen, Germany). The sludge was recirculated through the membrane by a peristaltic pump (Watson-Marlow 520s, Thermo Fisher Scientific, Göteborg, Sweden).

The permeate was extracted by a peristaltic pump (Watson-Marlow 120s, Thermo Fisher Scientific, Göteborg, Sweden). The pH was controlled by the provision of a pH electrode (Model sc1000 Controller, HACH Company, Loveland, CO, USA) and two peristaltic pumps (Watson-Marlow 1200s, Thermo Fisher Scientific, Sweden) for acid and base addition. The temperature in the AnMBR system was measured by a temperature sensor Pt1000 (Metrohm, Barendrecht, The Netherlands), and controlled by using a recirculating water bath (Thermo Haake DC 10, Thermo Fischer Scientific, Göteborg, Sweden). The amount of biogas produced by the AnMBR system was measured with a drum-type gas meter (Ritter, Boshum, Germany). Sample ports were provided to regularly monitor and control the sludge characteristics in the reactor. Figure 1 shows a schematic setup of the AnMBR systems.

### 2.3. Experimental Procedure

Both AnMBR systems were inoculated with crushed sieved (600 μm mesh size) granular sludge from a full-scale expanded granular sludge bed (EGSB) reactor (DSM, Delft, The Netherlands). The AnMBR systems were operated under mesophilic conditions (35 ± 1 °C). Both systems were fed with synthetic dairy wastewater by diluting 1/10th of whole milk. Additionally, nutrients and micronutrients were added to the system following the recipe of [19]. The synthetic influent wastewater exhibited the following average characteristics: 9988 ± 595 mg COD/L; 1760 mg FOG/L with 2.87 g COD/g FOG; 2089 ± 884 mg TSS/L; 170 ± 65 mg NH_4_^+^-N/L, and pH 5.8. Approximately 50% of the total COD contained in the wastewater were lipids.

During the first 69 operational days, both reactors were operated at an SRT of 30 days by wasting sludge at 330 mL/d. This period is further referred to as the coupled period. Hereafter, one of the reactors was operated at an SRT of 20 days (R20) by wasting 500 mL/d of sludge, and the other was operated at an SRT of 40 days (R40) by wasting 250 mL/d. The latter period is further referred to as the uncoupled period. The OLR was gradually increased by steps of 0.5 g/(L d) every 5 days, increasing the influent flowrate with steps of 0.5 L/d until an OLR of 4.7 ± 0.8 g/(L d) was reached for both systems. Once the final OLR was reached, the systems were operated at an influent flowrate of 4.7 ± 0.6 L/d, resulting in an HRT of 2.1 days for each reactor.

The sludge was entering the cross-flow membrane system at a flowrate of 38 L/h (912 L/d) to set a membrane cross-flow velocity of 0.5 m/s. Every 890 s of membrane filtration, a 10 s backwash was performed. A full-scale filtration membrane module was used to simulate full-scale operational conditions. Consequently, the provided membrane area was much larger than the required membrane area to process the influent wastewater flowrate at the target operational fluxes. Therefore, to operate at the desired fluxes while sustaining a constant level in the AnMBR vessel, a fraction of the permeate was directed to a 5 L permeate collection tank and from there recirculated to the AnMBR. The remaining permeate fraction (equal to the influent wastewater flowrate) was taken out of the system via the 5 L permeate tank using a peristaltic pump. The AnMBR systems were started at an operational flux of 2.5 L/(m^2^ h). After one week of operation, the flux was increased to 5 L/(m^2^ h); subsequently, the flux was increased by 5 L/(m^2^ h) increments until reaching an operational flux of 20 L/(m^2^ h) on operational day 60. After operating the system for approximately 60 additional days at this flux, the flux was reduced to 10 L/(m^2^ h) and remained at this level until the end of the experiment. Under the latter conditions, a permeate production of 11.8 L/d was obtained and approximately 4.7 L/d was taken out of the system, while the remaining 7.1 L/d was recirculated to the AnMBR vessel. Membrane chemical cleaning was performed when the TMP reached a value of 600 mbar. The chemical cleaning was started by performing a physical cleaning of the membrane to remove the cake layer; then, the membrane was soaked in a sodium hypochlorite solution (500 mg/L) for approximately one hour. After washing the membrane with clean water, the membrane was soaked in a 1% citric acid solution for approximately one hour. This cleaning procedure was provided by the membrane manufacturers.

### 2.4. Sludge Properties Determinations

The samples for determining the sludge properties were collected and analyzed as follows:

The TSS concentration was analyzed twice a week following the Standard Methods of APHA [20]. The sludge dynamic viscosity was determined twice a week with a viscometer (HAAK Viscotester 550, Thermo Fisher Scientific, Massachusetts, USA) at a shear of 700 rpm [7]. The PSD was determined every 20 days using a laser particle size analyzer (Microtrac S3500, Verder Scientific, Haan, Germany) with a diameter particle detection capacity range of 0.01 to 2800 µm. The median particle size (MPS) was defined by the central point of the peak of the PSD curve. This value was calculated and reported by the software controlling the particle size analyzer.

The capillarity suction time (CST) determination predicts the ability of the sludge to release water. A high CST value corresponds to a sludge difficult to dewater; thus, such sludge would be more difficult to filter through a membrane filtration system. The CST was determined once a week using a CST apparatus (Capillary Suction Timer 304 M, Triton Electronics, Essex, UK) and a standard paper filter (Whatman No. 17, Merck KGaA, Darmstadt, Germany) [7].

The SRF determination can provide information on the level of compaction of the cake layer [17]. This determination involves exposing the sludge to a dead-end filtration process. The time needed to produce a given volume of filtrate (permeate) is reported as the SRF [7]. Cake-layer formation has been reported as the most important fouling mechanism in AnMBRs [19]; therefore, the SRF determination can provide relevant information on the membrane filtration performance when operating the AnMBR system. The SRF was determined every two weeks using a dead-end filtration cell (Millipore 8050, Merck KGaA, Darmstadt, Germany) after the AnMBRs were set to their respective SRT. The samples were filtered under a constant pressure of 0.5 bar through a standard paper filter (Whatman Grade GF/F Filter 47 mm, Merck KGaA, Darmstadt, Germany) with a pore size of 0.7 µm. The SRF was calculated as described in Equation (1) [21]. The slope term (b) in Equation (1) was calculated as the slope of the filtration time as a function of the filtrate-volume (t/V) versus the filtrate-volume (V).
(1)$$\mathrm{SRF}\;=\frac{2\cdot \Delta P\cdot A^{2}\cdot b}{\mu \cdot C}$$
where:

SRF = Specific Resistance to Filtration (m/kg)

ΔP = Pressure of filtration (N/m^2^)

A = area of the filter paper (m^2^)

b = slope of the filtration time (s/m^6^)

µ = viscosity (N s/m^2^)

C *=* TSS concentration (kg/m^3^).

The SF determination can provide information on the membrane fouling potential due to the presence of fine particles and soluble compounds such as SMP and colloids. The SF determination evaluates the presence of such substances in the sludge supernatant. The presence of those compounds may lead to an increase in membrane fouling either due to a decrease in cake layer porosity and/or due to blocking of the membrane pores [12,14,15]. The SF was determined every two weeks after the AnMBRs were set to their respective SRT by centrifuging the sludge at 17,500× *g* for 10 min. Then, the supernatant was filtered through a 0.22 µm membrane filter (Whatman membrane filters nylon 47 mm, Merck KGaA, Darmstadt, Germany) in a stirred dead-end filtration cell (Millipore 8050, Merck, KGaA, Darmstadt, Germany) under a constant pressure of 0.5 bar [7].

The EPS refers to different classes of macromolecules such as proteins, carbohydrates, nucleic acids, and other polymeric substances secreted by the microorganisms; EPSc refers to the carbohydrate fraction, while EPSp refers to the protein fraction [12]. The SMP consists of proteins, carbohydrates, and other soluble cellular components present in the soluble fraction of the sludge mixture; similarly, SMPc refers to the carbohydrate fraction and SMPp refers to the protein fraction [12]. The EPS content was determined every two weeks after the AnMBRs were set to their respective SRT following the procedure by Dereli [7]. The EPS was extracted by thermal treatment at 100 °C for one hour. After the thermal extraction procedure, the sample was centrifuged at 17,500× *g* for 10 min, and then the sample was filtered using a 0.22 µm filter (Whatman UNIFLO 25 syringe filters, Merck KGaA, Darmstadt, Germany).

### 2.5. Total Resistance to Filtration

The total resistance to filtration can be defined as the ratio between the TMP and the dynamic viscosity and membrane flux. The total resistance to filtration includes all specific resistances to filtration, i.e., the intrinsic membrane resistance, the cake-layer resistance, and the resistance caused by pore blocking due to accumulating inorganic and/or organic compounds in the surface or the pores of the membrane. The total resistance to filtration (R_total_) was calculated following Equation (2).
(2)$$R_{\mathrm{total}}=\frac{\mathrm{TMP}}{\mu \;\cdot J}$$
where:

R_total_: total resistance to filtration (1/m)

TMP: transmembrane pressure (Pa)

µ: dynamic viscosity of water (Pa s)

J: flux (m^3^/(m^2^s)).

The different (individual) resistances to filtration as indicated in Equation (3) were determined when performing the cleaning in place (CIP) interventions according to Meng et al. [14]. The intrinsic membrane resistance (R_intrinsic_) was determined at the very beginning of this research when using the new (unused) membrane. The cake-layer resistance (R_removable_), the resistance caused by inorganic and organic foulants removed by chemical cleaning (R_irreversible_), and the resistance caused by foulants that cannot be removed by chemical cleaning (R_irrecoverable_) were determined after performing the CIP intervention. The membrane was first rinsed with clean water, and it was later cleaned with sodium hypochlorite and citric acid. After each cleaning intervention the resistances to filtration were determined and each of the individual resistances were calculated following Equation (3).
(3)$$R_{\mathrm{total}}=R_{\mathrm{intrinsic}}+{\;R}_{\mathrm{removable}}+{\;R}_{\mathrm{irreversible}}+{\;R}_{\mathrm{irrecoverable}}$$
where:

R_total_: total resistance to filtration (1/m)

R_intrinsic_: intrinsic membrane resistance (1/m)

R_removable_: cake layer resistance which can be physically removed by flushing with water (1/m)

R_irreversible:_ caused by inorganic and organic foulants that can be removed by chemical cleaning(1/m)

R_irrecoverable_: caused by foulants which cannot be removed by physical or chemical cleaning (1/m)

## 3. Results

The biological performance of the two reactors was thoroughly assessed and published in our previous work [22]. Results showed COD removal efficiencies of up to 99% in both AnMBR systems [22] and stable biogas production of 0.31 ± 0.02 NL CH_4/_(g COD removed) and 0.32 ± 0.02 NL CH_4_/(g COD removed) for R20 and R40 reactors, respectively. Moreover, an inhibitory effect of the biomass due to the presence of LCFA compounds was not observed. Apparently, at the evaluated process operational conditions, the AnMBR systems exhibited a much better biological performance treating lipid-rich dairy wastewater, compared to conventional anaerobic wastewater treatment systems [1,7]. Comparing both AnMBRs, the R40 reactor exhibited higher COD removal efficiencies, higher biogas production, and lower biomass specific lipid concentrations compared to the R20 reactor.

### 3.1. Physicochemical Sludge Properties

#### 3.1.1. Total Suspended Solids

Figure 2a describes the changes in TSS concentration during the entire operational period. The reactors were inoculated with sludge at a TSS concentration of approximately 16 g/L. During the coupled period, the two reactors were operated at the same SRT of 30 days; the TSS concentration initially decreased in both reactors until day 69 when the SRT was set to 20 and 40 days for R20 and R40, respectively. The TSS in both reactors slowly reached stable values at approximately (6.8 ± 0.3) g TSS/L and (12.4 ± 0.5) g TSS/L for the R20 and R40 reactors, respectively, meaning a steady-state operation. The TSS values obtained in this study were similar to the TSS concentrations reported in the literature [7,8,9].

#### 3.1.2. Dynamic Viscosity

Until day 99, the dynamic viscosity remained similar in both reactors, after which an increase was observed in both reactors (Figure 2b). At the end of the experimental period, the dynamic viscosity reached more or less 15 and 9 mPa.s in R40 and R20, respectively. In a study evaluating the rheological characteristics of anaerobic sludge, Pevere et al. [23] reported a dynamic viscosity of approximately 5–6 mPa.s, which is distinctly lower compared to the values found in our present research applying a similar shear rate.

#### 3.1.3. Mean Particle Size and Particle Size Distribution

Figure 2c shows the mean particle size (MPS) as a function of the exposure time for both reactors. A continuous decrease in the MPS was observed for both systems until a similar value was reached at the end of the evaluated period. Similar findings were also reported in literature [7]. The PSD shifted to lower particle size values in both reactors during the evaluated period (Figure 3a,b). Similar trends were observed regardless of the applied SRT in the reactor. The reduction in the particle size was attributed to the applied shear forces in the cross-flow filtration unit, and the use of peristaltic pumps for the feed-water recirculation.

### 3.2. Sludge Filterability Properties

#### 3.2.1. Capillary Suction Time

Figure 4a shows the CST of the sludge of both reactors for the entire evaluated period. An initial decrease in the CST values was noticed immediately after the inoculation of the reactors. However, from day 100 the CST values started to rapidly increase. From day 140 until the end of the experiment (day 189), average values of 956 ± 168 s and 1321 ± 138 s were found for R20 and R40, respectively. The specific or normalized CST, i.e., the CST divided by the TSS, for the entire evaluated period was also calculated. Similar to the CST values, after an initial drop, a steep increase in the specific CST was observed from day 100, which stabilized on day 140. In the final period, specific CST values of 138 and 112 s L/g for R20 and R40, respectively, were calculated.

Results in Figure 4a showed higher CST values for R40 compared to R20, indicating a better dewaterability and suggesting better filterability for R20 sludge compared to the R40 sludge. However, very similar and even slightly higher specific CST values were calculated for the R20 sludge compared to the R40 sludge, indicating the opposite trend. The specific CST values obtained for both reactors were distinctly higher than the values of 40–50 s L/g for anaerobic sludge in conventional anaerobic wastewater treatment systems, as reported in the literature [15,21,23,24].

#### 3.2.2. Specific Resistance to Filtration

Figure 4b shows the assessed specific resistance to filtration (SRF) values between days 69 and 160 for both reactors. It is important to highlight that the SRF is normalized to the TSS concentration in the sample. Therefore, SRF values provide an indication of the specific sludge filterability, similarly to the specific CST values previously discussed. The SRF steadily increased in both reactors, reaching maximum values of 12 × 10^14^ (m/kg) and 8 × 10^14^ (m/kg) on the operational day 140 for the R20 and R40 reactors, respectively. Dereli et al. [7] reported SRF values approximately one order of magnitude lower compared to the values shown in Figure 4b, thus indicating a worse sludge filterability in our present study. The high SRF values coincided with higher TMP values compared to the TMP values reported by Dereli et al. [7] while working at a similar operational flux. As observed with the specific CST values, the SRF results of the R40 sludge indicated a better filterability compared to the R20 sludge. This difference was even more noticeable in the SRF assessments compared to the specific CST assessments.

#### 3.2.3. Supernatant Filterability

Figure 4c describes the supernatant filterability (SF) for the two reactors in the experimental period from day 69 to 170. SF values of approximately 1.25 mL/min and 0.96 mL/min were found on day 70 for the R20 and R40 reactors when the SRT was set to 20 and 40 days, respectively. Hereafter, the SF decreased and stabilized from day 110 onwards, showing similar SF values in both reactors ranging from 0.2 to 0.4 mL/min with values similar to previous studies [7].

### 3.3. Presence of Soluble Substances in the Sludge Matrix

#### 3.3.1. Extra Polymeric Substances

Figure 5a presents the concentration of EPS in the sludge over time. Both the protein fraction of the EPS (EPSp) as well as the carbohydrate EPS fraction (EPSc) are presented. Up to day 100, concentrations of approximately 30 and 20 mg EPSp/g SS were reported for reactors R20 and R40, respectively. Hereafter, the EPSp concentrations increased, reaching maximum values on day 145 of 110 and 85 mg EPSp/g SS for R20 and R40, respectively. Then, the EPSp concentration slightly decreased towards the end of the experimental period. Lower values were reported for the EPSc compared to the EPSp for both reactors, showing EPSc concentrations below 20 mg/g SS throughout the experimental period. Results agree with the literature, showing EPSp concentrations ranging from 11 to 120 mg EPSp/g SS and EPSc concentrations ranging from 7 to 40 mg/g SS for aerobic MBRs and anaerobic up flow sludge bed filters [12]. The mentioned studies also reported an inverse relationship between the filtration performance and the concentration of EPSp and EPSc.

#### 3.3.2. Soluble Microbial Products

Figure 5b shows the SMPc and SMPp concentrations in reactors R20 and R40. Similar to the EPSp concentrations, the SMPp concentrations started to increase after day 100. On day 120, the SMPp concentrations reached maximum values of 425 and 300 mg SMPp/L for R20 and R40, respectively. Hereafter, the SMPp concentrations decreased to 150 and 125 mg SMPp/L on day 170 for R20 and R40, respectively. Le-Clech et al. [12], reported lower SMPp concentrations compared to our present study, ranging from 8 to 34 mg SMPp/L for aerobic and anaerobic MBRs operated at TSS concentrations ranging from 7 to 14 g/L. In addition, the authors reported a negative correlation between the SMPp concentration and the membrane filtration performance. The SMPc concentration for both reactors remained more or less constant during the entire experimental period, ranging from 50 to 100 mg SMPc/L. In addition, Le-Clech et al. [12] reported lower SMPc concentrations compared to our present study, i.e., ranging from 5 to 14 mg SMPc/L for aerobic and anaerobic MBR systems operated at TSS concentrations from 7 to 14 g/L, which are similar TSS concentrations, as in our present research. In addition, as observed with the SMPp, the authors reported a negative correlation between the SMPc concentration and the membrane filtration performance.

### 3.4. Membrane Filtration Performance

#### 3.4.1. Trans-Membrane Pressure and Flux Profile

Figure 6a shows the TMP values for both reactors at the different operational fluxes during the entire experimental period. Results clearly show that the TMP for the R40 reactor was consistently higher than the TMP for the R20 reactor (with some exemptions). The flux was increased stepwise until reaching a maximum flux of 20 L/(m^2^ h) on day 62, which coincided with a recorded TMP value of 71 and 107 mbar for R20 and R40, respectively, only showing a slight increase. A membrane CIP intervention was carried out on day 62 in an attempt to reduce the TMP. The CIP interventions are indicated in Figure 6a by a solid vertical green line. However, after the first CIP intervention the reactor operation continued to be operated at the target flux, resulting in a continued increase of the TMP reaching values of 301 and 455 mbar for R20 and R40, respectively, on day 100. The reported values were close to the maximum suggested operational TMP value of 500 mbar. Thus, two additional CIP interventions were carried out on days 85 and 110. In addition, the flux was reduced on day 95, first to 18 L/(m^2^ h) and subsequently a few days later to 15 L/(m^2^ h) expecting to reduce the observed TMP values. After the third CIP intervention on day 110, the TMP values stabilized for both reactors. However, approximately 10 days after, on day 120, the TMP values increased again, reaching similar TMP values as previously observed before the third CIP intervention. Thus, on day 130, the flux was reduced to 10 L/(m^2^ h) in both reactors to avoid potential damage to the membranes and to attain a stable membrane-filtration performance. Two additional CIP interventions were carried out on days 140 and 180, which did not result in major improvements regarding observed TMP values. In addition, as indicated in Figure 6a, throughout the experimental period the OLR was increased from an initial value of 1.0 g COD/(L d) until reaching the value of 4.7 g COD/(L d) on day 89. The increments in OLR are shown by dotted lines in Figure 6a. The increase in the OLR could have also impacted the biomass characteristics, probably contributing as well to the reported TMP values. The operational flux of 10 L/(m^2^ h) was slightly lower compared to the operational fluxes of 10 to 14 L/(m^2^ h) for AnMBR systems reported in the literature [7].

#### 3.4.2. Total Resistance to Filtration (R_total_)

Figure 6b shows the R_total_ for both reactors during the entire experimental period. At the start of the experiment, the R_total_ for the R20 reactor was slightly higher than for the R40 reactor, even though both reactors were operated at the same SRT of 30 days. Then, after day 62 when the SRT of 20 and 40 days was set for R20 and R40, respectively, the R_total_ increased continuously for both reactors. Since the R_total_ directly relates to the TMP, similar trends as observed in Figure 6a for the TMP were expected for the R_total_. After setting the operational flux at 10 L/(m^2^ h) at day 130, the R_total_ stabilized at average values of approximately (1.5 ± 0.3) × 10^13^ L/m and (1.8 ± 0.2) × 10^13^ L/m for R20 and R40, respectively. The reactor R40 exhibited a higher total resistance to filtration compared to the reactor R20. Similar total resistance to filtration values is reported in the literature [7].

The individual resistances contributing to the total resistance to filtration were calculated according to Equation (3). The results of the different individual resistances, calculated for days 140 and 188, are presented in Table 1 for both reactors. The cake-layer resistance exhibited the highest contribution to the total resistance to filtration for both reactors at the two evaluated operational days; this observation agrees with previous findings reported in the literature on AnMBRs [7,25,26]. Results show a higher contribution of the cake-layer total resistance in R40 compared to R20, which likely can be attributed to the presence of a thicker cake layer present in R40 than in R20.

## 4. Discussion

### 4.1. Physicochemical Sludge Properties

The TSS concentration apparently exerts an important role on membrane fouling, thus affecting the attainable membrane flux [3,11,27]. A higher SRT implies a higher TSS concentration in the reactors. The TSS concentrations in R40 and R20 were 12.4 ± 0.5 g TSS^/^L and 6.8 ± 0.3 g TSS/L, respectively. The higher TSS concentration in R40 coincided with higher TMP values and higher R_total_ than in R20, indicating a worse membrane-filtration performance. A decrease in the membrane-filtration performance was observed at higher SRT as reported in a previous study [28]; beyond a critical TSS concentration of approximately 10 g TSS/L, the sludge filterability and membrane-filtration performance deteriorated.

A high SRT condition, such as in R40, will result in high total-membrane resistance and low filtration performance. In addition, increased TSS concentrations concomitantly result in an increased presence and concentration of viscous substances in the sludge, such as carbohydrates and proteins [14], resulting in higher dynamic viscosities, as was observed in R40 [21,29].

The PSD has an important role in membrane fouling [30], largely determining membrane-filtration performance. MBRs equipped with cross-flow membrane filtration units are characterized by high shear stress on the sludge. Such shear forces promote sludge flocs breakage with a consequent change in the PSD of the sludge. The PSD shifts to the lower particle-size range, reducing the number of large particles and increasing the number of small particles, enhancing membrane fouling. In this research, the AnMBR systems were operated at a relatively low cross-flow velocity of 0.5 m/s [7,8,31]; however, it was apparently large enough to shift the PSD of the sludge to smaller particle sizes at the two evaluated conditions in R20 and R40. Although slightly higher MPS were observed for the R40 reactor compared to R20, the changes in PSD and MPS observed in this study were very similar for the two evaluated reactors; thus, the evaluated SRTs and related sludge concentration seemed not to play a significant role on promoting or avoiding such a shift in PSD. Similar shear forces were applied to both reactor; thus, similar patterns regarding the PSD were expected.

In the present study, the MPS decreased in both reactors (R20 and R40), while the TMP and R_total_ increased in both systems. The deterioration in the membrane-filtration performance as a function of the operational time is likely due to multiple factors, including the increase in TSS concentration, increase in dynamic viscosity, and sludge characteristics, among others [14,15,25,26,32,33].

However, the sludge particle size could have had a more pronounced impact on the membrane filtration performance in our present study compared to other causes. Various factors are presumed to be inter-related; for instance, the higher concentration of TSS solids in R40 also may have contributed to a larger amount of fine-sized particles in R40 than in R20. Thus, both the shift on the PSD to lower particle sizes, as well as the presence of large TSS concentrations, very likely increased synergistically the observed TMP and R_total_ in R40 compared to R20.

A larger concentration of TSS may directly affect the membrane-filtration performance by promoting the formation of a thick cake layer at the surface of the membrane. Moreover, the TSS concentration may also impact the other sludge properties such as the sludge dynamic viscosity, sludge PSD, CST, and presence of EPS and SMP compounds, among others. The impact of these properties on the membrane filtration performance is discussed below.

### 4.2. Sludge Filterability Properties

The CST values increased in both reactors as a function of the operational time; however, higher CST values were reported for the R40 reactor compared to the R20 reactor. High CST values indicate a poor filterability of the sludge [6]; thus, the results obtained in this study clearly indicate a better sludge filterability for the sludge in the R20 reactor. However, when normalizing the CST to the TSS concentration, similar specific CST values were found for both the R20 and R40 reactors; even higher specific CST were reported for the R20 sludge than for the R40. That is, both the R20 and R40 sludge exhibited very similar sludge dewaterability properties. The non-normalized CST resembles the differences in the filtration resistance in both reactors [34]. The increase in the CST values as a function of the operational time were associated with an increase in TMP and R_total_. Thus, the TSS concentration was seemingly the major cause affecting membrane filtration performance, rather than the filterability of the sludge. Dereli et al. [7] reported similar CST values when operating an AnMBR treating lipid-rich wastewater (corn-based thin stillage). The lipid-rich influent wastewater could eventually negatively affect the sludge filterability; nonetheless, the sludge concentration rather than the sludge characteristics (specific CST) seemed to pose the major negative effects on the membrane-filtration performance [35] under the given hydraulic conditions. As a result, even though the CST in R40 had been higher than the R20, when it was normalized the CST/TSS had similar values for both reactors.

Similar findings as for the specific CST were observed for the SRF. The SRF is also indicative of the sludge filterability [14,15]; the SRF is normalized to the concentration of suspended solids in the sample. Higher SRF values indicate a more compact (less permeable) sludge cake layer. The SRF is normalized to the TSS concentration; therefore, it is a less predictive indicator of the actual membrane filtration performance.

The specific sludge filtration characteristics for the R40 sludge were better than for the R20 sludge; however, the cake layer in the R40 was thicker—twice as thick—resulting in a worsened filtration performance. Increasing the cross-flow velocities on the membrane surface could have eventually counteracted such filtration loss, which however also could also result in an increased amount of fine particles, worsening the sludge filterability.

The decrease in SF values could be explained by the presence of colloidal material remaining in the supernatant after centrifuging the sludge for carrying out the SF determination. Apparently, the differences in TSS concentrations and the possible larger fine-particles concentration in the R40 reactor did not yield different SF values under the given hydraulic conditions [36]. Thus, the SF does not provide major information on the membrane-filtration performance for the evaluated conditions.

### 4.3. Presence of Soluble Substances in the Sludge Matrix

Higher specific EPSp values were observed for the R20 reactor compared to the R40 reactor. Thus, the sludge in the R20 reactor may exhibit a worse filterability due to the presence of specific higher concentration of EPSp compared to the sludge in R40. As previously explained for the specific CST and SRF, it could eventually occur that the sludge in the R20 reactor produced higher specific amounts of EPSp. Even though the TSS specific EPSp concentration in R20 was higher compared to R40, the absolute concentration of EPSp was higher in R40 compared to R20, possibly explaining the worse membrane-filtration performance observed in R40 compared to R20. Similar findings could be observed for the EPSc content; however, the specific EPSc concentrations were very similar for the R20 and R40 reactor. The absolute concentration of EPS in the sludge increases with the experimental period. The changes in PSD could be indicative for the release of EPS substances originally present in the sludge. Consequently, the shift in PSD may explain the increase in absolute EPS concentration, contributing to a worsened membrane-filtration performance of the R40 reactor compared to the R20 reactor.

Similar TSS specific SMPp and SMPc concentrations were observed in both the R20 and R40 reactors as in the literature [37]. A subtle increase in the SMP values was followed by stable SMP concentrations at the end of the evaluation. However, the TMP and R_total_ deteriorated as the time progressed. Therefore, the observed differences in the absolute SMP concentrations between the two reactors could have contributed to the differences in the observed membrane-filtration performance between the two systems.

### 4.4. Membrane Filtration Performance

The reactor R40 exhibited higher TMP values compared to the R20 reactor at the same operational fluxes during the entire evaluated period. The observed TMP values in this evaluation for both reactors were higher than in the literature [7,38]. Particularly, Dereli et al. [7] also treated a lipid-rich wastewater (palm oil mill effluent (POME)) in an AnMBR and reported lower TMP values, from to 100 to 200 mbar. The authors used a similar experimental setup as in this research. However, the influent wastewater, although lipid-rich, was characterized by a different FOG content and LCFA profiles compared to the influent wastewater in our research, i.e., the percentage of palmitic acid in POME is approximately 42% of the total LCFA compared to 21% found in the whole milk used in our study [1]. Therefore, the observed differences in TMP values might be attributed to the different characteristics of the obtained sludge treating different types of wastewaters, i.e., corn-based bioethanol thin stillage wastewater versus synthetic dairy wastewater. Additionally, although both wastewaters are rich in lipids, they also showed different lipid profiles and different concentrations of other substances such as divalent ions (calcium) [15,31,39] and colloidal material [14], eventually leading to a different filtration performance.

When looking at the various sludge-filterability indicators, i.e., parameters describing the inherent ability of the sludge to lose water, such as the specific CST and the SRF, no major differences were found between the R20 and R40 sludges. Moreover, in some cases, the R40 sludge showed a slightly better sludge filterability. Similar observations were made regarding the presence of fouling compounds such as EPS and SMP. Moreover, in a previous study carried out by the authors [22], the sludge specific lipids and LCFA concentrations were higher in the R20 reactor compared to the R40. That is, in terms of sludge filterability, the R40 sludge exhibited lower specific lipid and LCFA concentrations, eventually showing a better inherent filterability. Therefore, the membrane performance in the two reactors seemed to be determined by other parameters beyond the inherent filterability of the sludge in each reactor. The concentration of TSS was eventually the most relevant physicochemical sludge parameter affecting the membrane filtration performance. The TSS concentration is one of the most important factors affecting the cake-layer formation and its thickness. The higher the SRT is, the higher the TSS concentration is and the thicker the cake-layer is during filtration, explaining the worse membrane-filtration performance of R40 compared to R20. In addition, the shift in PSD introduces larger concentrations of small particles, and this was more pronounced in the R40 reactor, owing to the increased TSS concentration. A large number of fine particles, together with the presence of a high overall TSS concentration, could have contributed to form a thicker and possibly more compact cake layer on the surface of the membrane, reducing the membrane-filtration performance. The increased TSS concentrations in R40 could additionally have led to increase the absolute concentrations of EPS and SMP substances, thus even contributing more to the cake layer.

In our previous work [22], a better overall biological performance was reported when operating the reactor at an SRT of 40 days compared to an SRT of 20 days. Higher organic matter conversion, higher biogas production, and lower sludge wastage were observed when working at an SRT of 40 days (R40) compared to an SRT of 20 days (R20). The results of our present research indicated that regarding the membrane filtration performance, the operation of the reactors at an SRT of 20 days is preferred. However, very likely, the filtration performance when operating at a high SRT can be further improved by introducing changes in the membrane operational strategy. Cake-layer formation and consolidation is the most important parameter affecting the total membrane resistance. Therefore, changes in the membrane operational strategy can eventually be proposed to reduce the impact of the cake-layer resistance on the overall membrane-filtration resistance [40]. Changes in the membrane operational strategy may include changes in the backwash frequency, the addition of a membrane relaxation period, changes in the cross-flow velocities, and introducing more frequent CIP interventions, among others. However, the effects of cleaning the membrane operational strategy on the membrane-filtration performance was not carried out as a part of this study; therefore, additional research would be needed to confirm such a hypothesis.

## 5. Conclusions

The SRT exerted an important effect on the sludge properties, including on the TSS concentration, dynamic viscosity, PSD, MPS, CST, SRT, and on the presence and concentration of EPS, SMP, lipids, and LCFA. The changes in sludge properties resulted in different membrane-filtration performances. The higher the SRT was, the worse the membrane-filtration performance was.The major individual contributor to the total resistance to filtration is the cake-layer resistance negatively affecting the membrane-filtration performance.The TSS concentration was the most important parameter determining the cake-layer resistance; thus, the membrane-filtration performance under the given evaluated process conditions including the membrane-cleaning regime.The changes in the SRT did not affect the specific sludge filterability.An SRT of 20 days resulted in a better membrane-filtration performance compared to an SRT of 40 days, attributed mostly to the lower sludge concentration in the reactor.

## Figures and Tables

**Figure 1 membranes-12-00262-f001:**
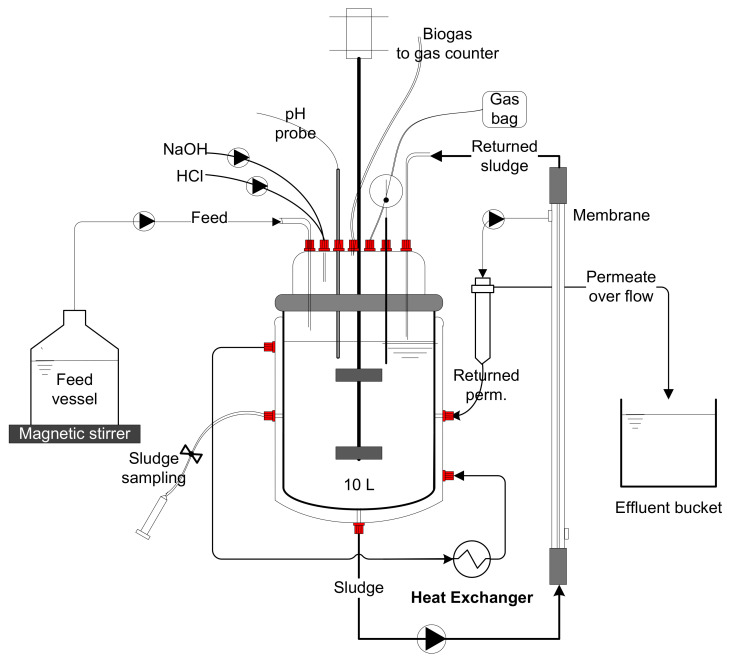
Anaerobic membrane bioreactors (AnMBRs) experimental setup.

**Figure 2 membranes-12-00262-f002:**
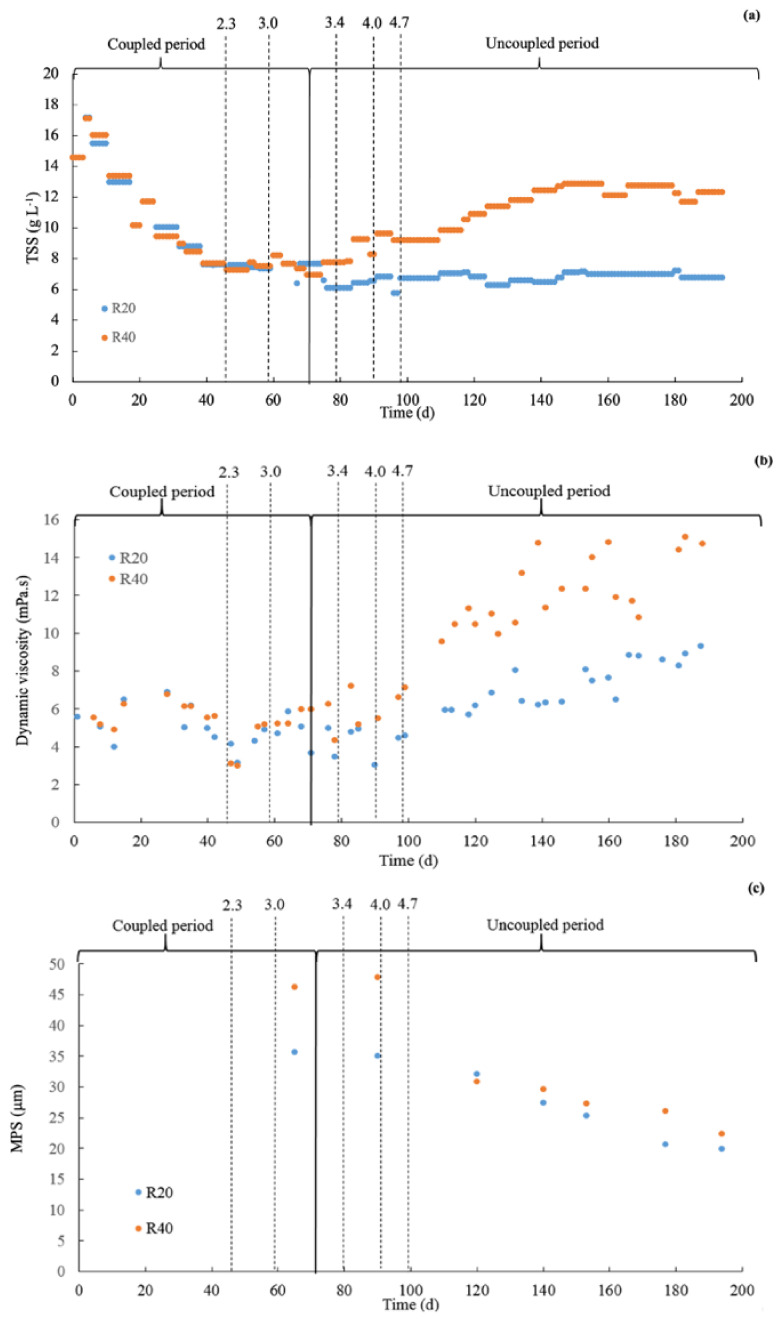
(**a**) Total suspended solids (TSS) concentrations, (**b**) Dynamic viscosity, and (**c**) Median particle size (MPS) over time for the R20 and R40 reactors. The dotted lines indicate the applied organic loading rate (OLR) (g COD/(L d).

**Figure 3 membranes-12-00262-f003:**
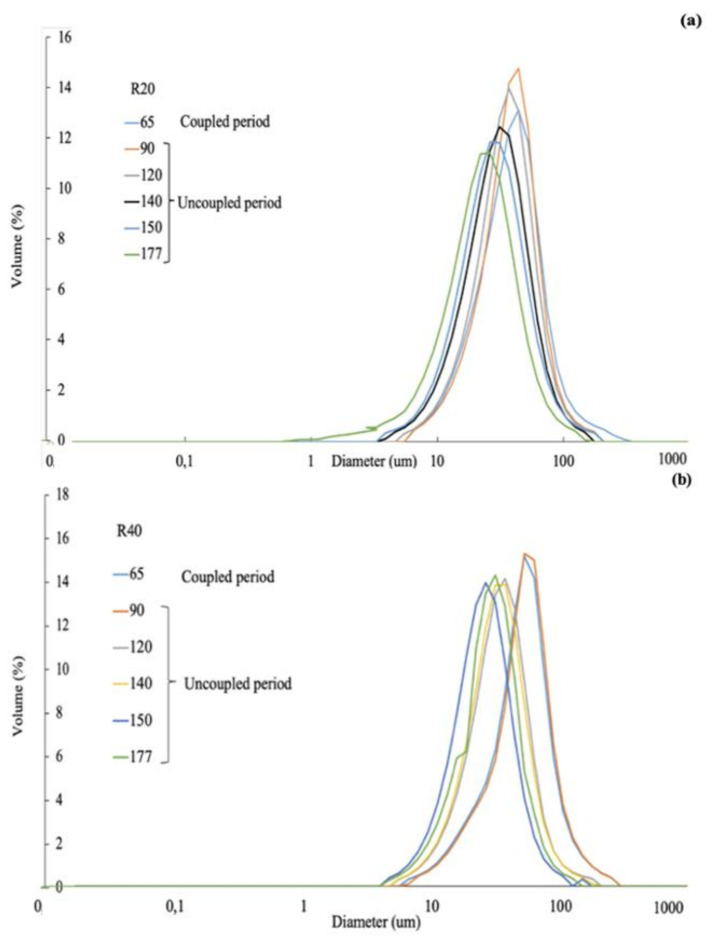
Particle size distribution (PSD) over operational time for (**a**) R20, and (**b**) R40 reactors over time.

**Figure 4 membranes-12-00262-f004:**
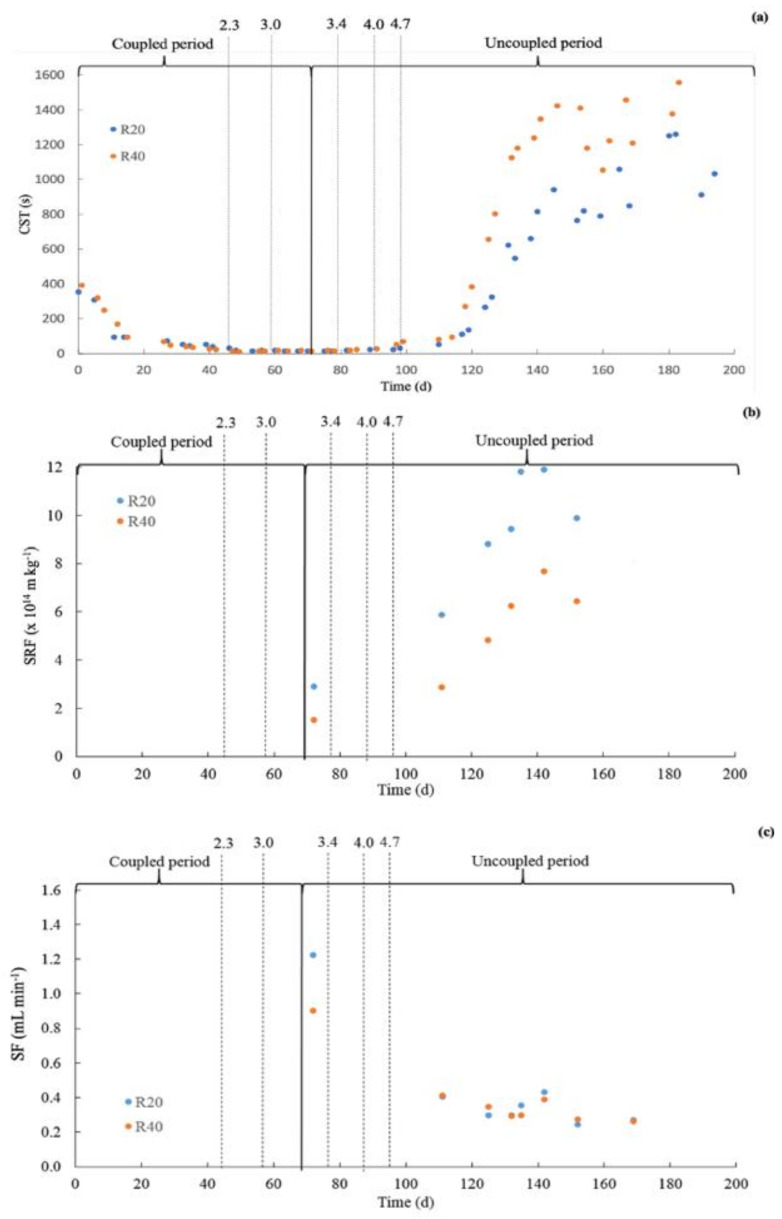
(**a**) Capillary suction time (CST), (**b**) Specific resistance to filtration (SRF), and (**c**) supernatant filterability (SF) for the R20 and R40 reactors over time. The dotted lines indicate the applied OLR (g COD/(L d)).

**Figure 5 membranes-12-00262-f005:**
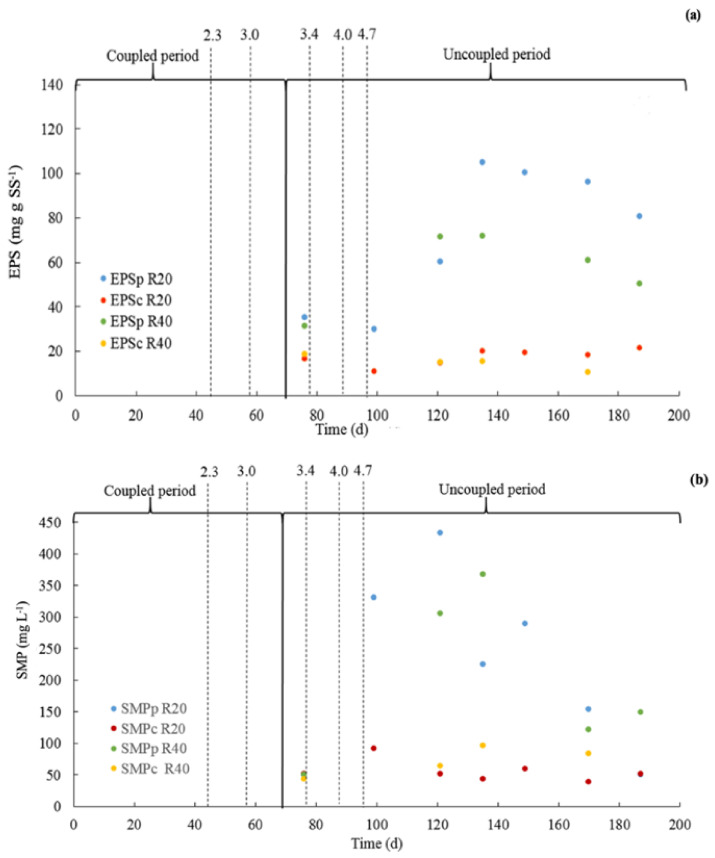
(**a**) Extracellular polymeric substances (EPS) fractions, and (**b**) Soluble microbial products (SMP) fractions for the R20 and R40 reactors over time. The dotted lines indicate the applied OLR (g COD/L d).

**Figure 6 membranes-12-00262-f006:**
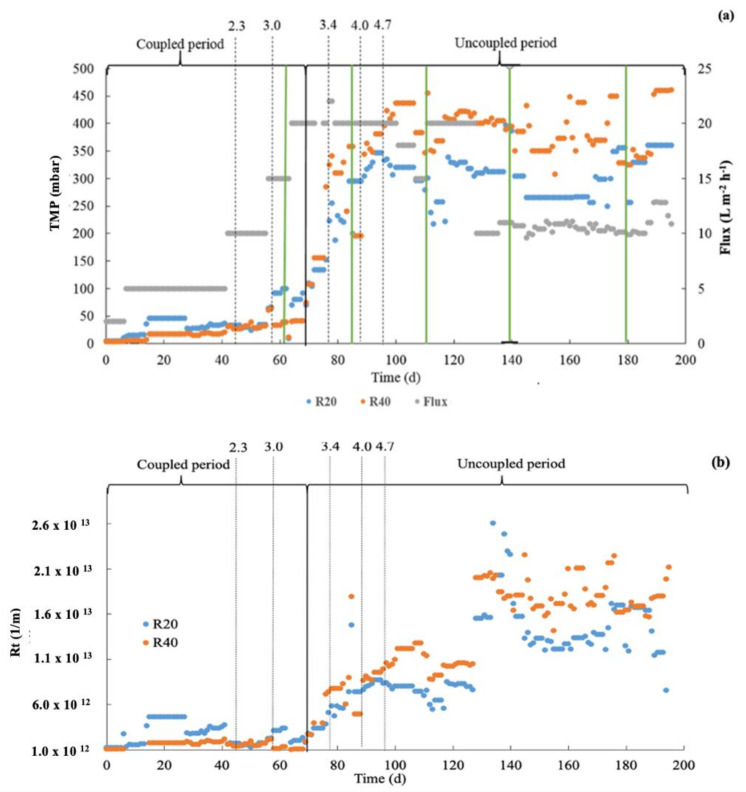
(**a**) Flux (L/m^2^ h) and transmembrane pressures (TMP) (mbar), and (**b**) Total Resistance to Filtration (R_tota_l) values over time for the R20 and R40 reactors. The dotted lines indicate the applied OLR (g COD/L d), and the green lines indicate the cleaning in place (CIP) interventions carried out in the reactors.

**Table 1 membranes-12-00262-t001:** Individual resistances for the R20 and R40 reactors at operational days 140 and 188.

Reactor	Operational Day	R intrinsic (%)	R Removable (%)	R Irreversible (%)	R Irrecoverable (%)
R20	140	2	75	21	2
	188	3	63	28	6
R40	140	2	78	18	2
	188	3	79	16	2

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
