# Peer review of "Influence of the Sludge Retention Time on Membrane Fouling in an Anaerobic Membrane Bioreactor (AnMBR) Treating Lipid-Rich Dairy Wastewater"

_membranes, 2022, doi:10.3390/membranes12030262_

Round 1

Reviewer 1 Report

The authors reported a study that compares the AnMBR performance at two different SRTs (20 and 40 days). In general, the authors provide a comprehensive discussion in this study. However, some issues should be soluted before publication:

  1. The introduction should be more concentrated and the novelty of this study should be highlighted.
  2. Line 80. Revise “says” to “days”.
  3. Membrane characterization such as SEM images are suggested to support the statements in this study. Especially, page 15 line 493-499, to support the statement about the cake layer is the most important factor to influence the total resistance, membrane characterization should be required.
  4. The authors are suggested to rewrite the discussion section. The discussion should be based on the authors’ results in this study rather than using results or viewpoints from literature. There are too many citations and viewpoints from other literature. For example, Line 510 “Lousada-Ferrira et al reported”, “Jeison and van Lier concluded that”, “Jeison and van lier also discussed”, “Khongnakorn et al reported that”, “Wu et al reported”…
  5. In the abstract and conclusion, the authors state that optimizing the membrane filtration operation (cleaning strategy, cross-flow velocities, duration of cycles, among others) may eventually contribute to operate the AnMBR system at higher SRT values and TSS concentrations without negatively affecting the membrane filtration performance. However, there are very limited results from this study to support this conclusion. Most of the discussion are based on viewpoints from other researchers or previous studies. The authors are suggested to revise the statement.

Reviewer 2 Report

See attachment

Round 2

Reviewer 1 Report

The authors revised the manuscript very well. I suggest publishing this version.